# The Childhood Opportunity Index 2.0: Factor Structure in 9–10 Year Olds in the Adolescent Brain Cognitive Development Study

**DOI:** 10.3390/ijerph22020228

**Published:** 2025-02-06

**Authors:** Julia C. Harris, Isabelle G. Wilson, Carlos Cardenas-Iniguez, Ashley L. Watts, Krista M. Lisdahl

**Affiliations:** 1Department of Psychology, University of Wisconsin-Milwaukee, Milwaukee, WI 53211, USAmedinak@uwm.edu (K.M.L.); 2Department of Population and Public Health Sciences, Keck School of Medicine, University of Southern California, Los Angeles, CA 90032, USA; 3Department of Psychological Sciences, Vanderbilt University, Nashville, TN 37212, USA; ashley.watts@vanderbilt.edu

**Keywords:** psychopathology, child opportunity index, neighborhood, adolescence

## Abstract

The built physical and social environments are critical drivers of child neural and cognitive development. This study aimed to identify the factor structure and correlates of 29 environmental, education, and socioeconomic indicators of neighborhood resources as measured by the Child Opportunity Index 2.0 (COI 2.0) in a sample of youths aged 9–10 enrolled in the Adolescent Brain Cognitive Development (ABCD) Study. This study used the baseline data of the ABCD Study (*n* = 9767, ages 9–10). We used structural equation modeling to investigate the factor structure of neighborhood variables (e.g., indicators of neighborhood quality including access to early child education, health insurance, walkability). We externally validated these factors with measures of psychopathology, impulsivity, and behavioral activation and inhibition. Exploratory factor analyses identified four factors: Neighborhood Enrichment, Socioeconomic Attainment, Child Education, and Poverty Level. Socioeconomic Attainment and Child Education were associated with overall reduced impulsivity and the behavioral activation system, whereas increased Poverty Level was associated with increased externalizing symptoms, an increased behavioral activation system, and increased aspects of impulsivity. Distinct dimensions of neighborhood opportunity were differentially associated with aspects of psychopathology, impulsivity, and behavioral approach, suggesting that neighborhood opportunity may have a unique impact on neurodevelopment and cognition. This study can help to inform future public health efforts and policy about improving built and natural environmental structures that may aid in supporting emotional development and downstream behaviors.

## 1. Neighborhood Quality

According to the American Community Survey in 2015–2019, 11.3% of the available United States census tracts (more than 73,000 tracts) had poverty rates of 20% or more for those under the age of 18 years [1]. Furthermore, more than 9% of the United States population was living in persistent poverty census tract locations. One key consequence of poverty is neighborhood quality, a social driver of health that pertains to the built environment children grow up in (e.g., access to quality education, resources for education, access to healthy food, a clean and walkable physical environment). Neighborhood quality and related socioeconomic factors are predictive of child health outcomes, including mental health [2] and brain [3] and cognitive development [4,5,6,7]. Furthermore, late childhood into preadolescence is also a time of shifts in independence and increasing autonomy as youths start spending more of their time outside the family context to engage with their peers and school systems in the community. Further, preadolescents spend a larger proportion of their time within their neighborhoods or areas close to their homes [8]. Research is needed using a large, nationally diverse sample to investigate co-occurring exposures that may differentially impact each other to contribute to child and adolescent development.

## 2. Rationale to Investigate Aspects of the Built and Natural Environment over and Above Socioeconomic Status

Environmental neuroscience research has focused on indicators of socioeconomic status (SES), such as household-level income and parental education, or race and ethnicity as indirect or proxy variables for neighborhood opportunity. A growing body of research is investigating the built and natural environment to expand the field’s focus to specific and unique neighborhood markers related to adolescent mental health and cognition [9,10,11], including proximity to major roads [12,13], toxic substances (e.g., lead or air pollution) [14,15,16], and extreme temperature [17]. Household SES, or individual-level income and educational attainment, may indicate the degree of caregiver support and access to resources within the home [18], whereas neighborhood-level factors include environmental and social influences (e.g., noise, air, lead pollution) as well as built characteristics of the environment (e.g., interactions with the community, access to quality education). Evidence is mixed when understanding the contributions of neighborhood factors (neighborhood socioeconomic composition) versus individual contextual factors (e.g., household income) to mental health. While some studies have identified the impact of individual-level factors on aspects of child mental health, household income alone may not be protective against individuals who are living in neighborhoods with lower poverty and disadvantage and the associated consequences of a low-resourced neighborhood [19]. Disentangling the neighborhood-level factors commonly associated with SES is crucial in understanding how the environment contributes to or promotes child and adolescent mental health development.

## 3. Rationale to Investigate Psychopathology and Behavioral Endophenotypes

It is becoming increasingly important to investigate the mechanisms that give rise to the associations between neighborhood quality and adolescent psychopathology, including internalizing and externalizing symptoms and related phenotypes, including impulsivity and behavioral activation. For instance, research suggests that the lack of neighborhood opportunity, socioeconomic disadvantage, or urban social stressors (e.g., urbanicity) may contribute to increased sensitivity in physiological stress systems, including the hypothalamic–pituitary adrenal (HPA) axis, among youths and adults [20,21,22]. Also, reduced neighborhood opportunity may promote mental fatigue, which may contribute to the depletion of top-down direct or voluntary attentional control [23,24,25], poorer self-control, and response inhibition among youths [26,27]. Alternatively, aspects of neighborhood opportunity that lend themselves to more compatible and adaptive environments (e.g., green space, lower noise or air pollution, etc.) may provide a restorative effect that may reduce stress and reduce depressive and anxiety symptoms [28,29,30]. Additionally, given that the rapid brain development and significant maturation of the hypothalamic–pituitary adrenal (HPA) axis contribute to increased stress-induced hormonal responses [31,32], investigating the relationship between neighborhood quality and adolescent psychopathology remains critical.

## 4. Links Between the COI and Psychopathology

The Childhood Opportunity Index (COI) uses geocoded data linked at the census tract level to provide scores on 29 neighborhood-level indicators of opportunity based on neighborhood data compiled from public sources (e.g., the U.S. Census Bureau, U.S. Environmental Protection Agency, National Center for Education Statistics). The COI has been investigated in the context of numerous health care access and physical health outcomes [33,34,35]. One study found that lower overall COI scores, indicating lower opportunity, were linked with higher cortisol [36]. However, few studies have investigated the COI and overall neighborhood quality factors and behavioral outcomes among youths. One study did not find a significant relationship between overall COI scores and anxiety and depression scores in a sample of adolescents aged 13–16 years old [37]. Another study using the Adolescent Brain Cognitive Development (ABCD) Study sample at baseline (youths aged 9–10) examined both the Area Deprivation Index (ADI) quintiles and the COI quintiles to examine relations with Child Behavior Checklist (CBCL) scores [38]. Greater neighborhood resources were associated with fewer externalizing symptoms, but adjusting for child- and family-level factors rendered the association between the COI and externalizing nonsignificant. However, research is needed to investigate the COI 2.0 beyond quintiles and composite scores to identify if different aspects of opportunity differentially associate with child psychopathology.

Using the ABCD baseline data, Xiao and colleagues (2023) used several neighborhood composite measures, including the COI, Social Vulnerability Index (SVI), and ADI to categorize neighborhood factors into four patterns: affluent community, high-stigma environment, high socioeconomic deprivation, and high crime and drug sales [39]. This study found that living in affluent communities was associated with fewer internalizing problems, externalizing problems, social problems, thought problems, and attention problems and total Child Behavior Checklist (CBCL) scores. Additionally, those living in high socioeconomic deprivation had the most severe internalizing problems, externalizing problems, social problems, and CBCL total score. While this study is particularly strong in identifying patterns of neighborhood opportunity, research is still needed to identify patterns within the COI within this age group and to extend to endophenotypes of adolescent psychopathology including impulsivity and behavioral approach systems.

## 5. Rationale for Factor Analysis of the COI 2.0 Among Youths in the ABCD Study

The COI 2.0 provides an overall summary score and three domain scores: (1) education, (2) health and environment, and (3) social and economic. In previous research, the overall summary score and domain scores have been associated with measures of mental health and cognitive functioning [40,41]. Importantly, this research links overall composite measures of neighborhood quality with neurodevelopment; however, identifying a broader range of neighborhood relationships may disentangle the constructs of economic, social, and health domains and offer more specificity in this age range to target youths as they age into adolescents. The Adolescent Brain Cognitive Development (ABCD) Study presents a unique opportunity to explore social drivers of the built environment among a large, socio-demographically diverse sample within the United States across various regions and with varying access to neighborhood resources. Importantly, the ABCD data contain a wide variety of social drivers of health-related variables, which allows researchers to address mechanisms that stem from proxy variables such as race, ethnicity, and general socioeconomic variables (e.g., parent education and income) [42,43].

Researchers have selected data-driven approaches, such as exploratory factor analysis, to investigate dimensions and to maximize the amount of variance explained in the data. Exploratory factor analysis can be utilized as a data reduction technique when given a larger set of observed variables (e.g., indicators of the COI 2.0), reducing data into unobserved latent variables, or factors [43]. Thus, research is needed to provide insights on the precise mechanism or causal pathway of neighborhood quality and opportunity that go beyond what a summary score or domain score can offer. Crucially, the importance of reducing the COI 2.0 into a subset of factors is two-fold: (1) to explore and validate the structure of the COI 2.0 among a wide sample of youths enrolled in the ABCD Study and (2) to enhance specificity in identifying targets of intervention in the environment. In this study, we used the ABCD Study baseline data to characterize dimensions of neighborhood quality using the COI 2.0 and examined these factors’ associations with important external criteria, including child psychopathology and allied personality traits (e.g., impulsivity, behavioral approach).

## 6. Materials and Methods

### Participants

The study used baseline data collected from the Adolescent Brain Cognitive Development (ABCD) Study, a diverse, national, prospective, longitudinal study that recruited 11,878 youths (age = 9–10 years old; *n* = 9767) [44,45,46] (see Table 1). Participants were not eligible to participate in the ABCD Study if they were not fluent in English, or had an MRI contraindication, a major neurological disorder, a gestational age of less than 28 weeks or a birth weight of less than 1200 g, birth complications that resulted in hospitalization for more than one month, uncorrected vision, or a current diagnosis of schizophrenia, an autism spectrum disorder (moderate, severe), an intellectual disability, or an alcohol/substance use disorder at the time of potential enrollment.

At baseline, the youth and one caregiver completed one to two in-person sessions, in which they completed a battery of assessments including the domains of mental and physical health [47], substance use [48], and peers, family, culture, and environment [49] and MRI scans [50,51]. The current study used data from participants with complete data from the demographic surveys [47], residential history [11,42], and youth- and caregiver-reported psychopathology and allied traits (e.g., impulsivity, behavioral approach; *n* = 9767) as part of Curated Release 5.1 (https://doi.org/10.15154/z563-zd24); thus, missing data were not random.

## 7. Measures

### 7.1. Neighborhood Quality

The ABCD Study provides scores from linked external datasets, including the Child Opportunity Index (COI 2.0) [52], which consists of the overall composite, three domain indices (economic, education, and health/environment), and the 29 indicators that make up the indices. The COI 2.0 indicators include proximity to high-quality early education centers, school poverty rate, access to healthy food, exposure to gray space, average airborne microparticle concentration, poverty rate, third-grade reading/math proficiency, and walkability, among other variables measuring the economic, education, and health domains. Indicator data were primarily sourced from various public datasets. For example, in the education domain, adult educational attainment is operationally defined as the percentage of adults aged 25 and over with at least a college degree, and these data were sourced from the American Community Survey. In the health and education domain, access to health food is defined as the percentage of households without cars that live over half a mile away from the nearest grocery store, and these data were sourced from the United States Department of Agriculture. In the social and economic domain, poverty rate is defined as the percentage of individuals living in households with an income below 100% of the poverty line, and these data were sourced from the American Community Survey (see COI 2.0 Technical Documentation [52]). The COI 2.0 data were linked to participant residential address data collected during the baseline visit (children aged 9–10 years) [53]. The Linked External Database Environment and Policy Working Group (LED Working Group) [11] identified the latitude and longitude of the residential addresses and linked the baseline residential geocodes to the external COI 2.0 dataset, which provides aggregated socio-demographic information at the census tract level (spatial data aggregations designed to have 4000 people in each [54]).

The COI 2.0 includes measures of early childhood education (e.g., third-grade math proficiency), educational and social resources (e.g., school poverty rate), healthy environments (e.g., access to healthy food, exposure to gray space), toxic exposures (e.g., average airborne microparticle concentration), and economic and social resources (e.g., poverty rate). Unlike other indices of neighborhood disadvantage (e.g., the Area Deprivation Index, ADI), the COI was developed to include factors specifically linked to healthy childhood development. The COI 2.0 was selected for geocoding in favor of the COI 1.0 mainly because it expanded the available dataset from only the 100 largest metro areas (47,000 census tracts) to nearly all U.S. census tracts (>72,000 census tracts), among other updates. Further, the included COI 2.0 measures most closely overlapped with the baseline collection year of the ABCD Study (2016–2018).

### 7.2. External Criteria

Youth Psychopathology. We used the Child Behavior Checklist, a well-validated self-report measure administered to the parents of youths investigating psychopathology signs and symptoms among youths [55]. Here, we used the externalizing symptoms, internalizing symptoms, attention problems, and thought problems scales. Externalizing symptoms include aggression or rule-breaking behavior. For example, items include “cruelty, bullying, or meanness to others” or “gets in many fights”. Internalizing symptoms include sadness, depression, anxiety, and loneliness. For example, items include “feels worthless or inferior” or “complains of loneliness”. Thought problems include obsessive thoughts, self-harm, and hallucinations. For example, items include “deliberately harms self or attempts suicide” or “sees things that aren’t there”. Attention problems include difficulty concentrating, daydreaming, and impulsivity. For example, items include “can’t sit still, restless, or hyperactive” or “daydreams or gets lost in their thoughts”. For a thorough description of the psychometric properties, see Achenbach, 2009 [56], and Barch et al., 2018 [47].

Behavioral Inhibition and Activation. We used a modified, 20-item version of the Behavioral Inhibition System (BIS) and the Behavioral Approach System (BAS) scales [57,58,59,60]. The BIS represents a psychological mechanism that promotes avoidance and inhibits behaviors to avoid negative consequences. For example, items include “I worry about making mistakes” or “I am hurt when people scold me or tell me that I do something wrong”. The BAS relates to pursuing a reward or positive reinforcement to promote reward-seeking behaviors. For example, items include “I crave excitement and new sensations” or “I am always willing to try something new, when I think it will be fun”. The BIS scale contains 7 items, and the BAS contains 13 items that make up three subscales (Drive, Reward Responsiveness, and Fun Seeking). The Drive subscale assesses goal motivation, the Reward Responsiveness subscale assesses sensitivity to pleasant reinforcers, and the Fun Seeking subscale assesses motivation toward novelty. Youths responded to each item on a 4-point Likert scale that ranged from 1 (very true for me) to 4 (very false for me). For a description of the psychometric properties, see Carver and White, 1994 [61], and Barch et al., 2018 [47].

Impulsivity. We used a 20-item modified version of the Urgency, Premeditation, Perseverance, Sensation Seeking, and Positive Urgency (UPPS-P) scale, which measures five specific facets of trait impulsivity [23,24,62,63]: Negative Urgency, or the tendency to act rashly when experiencing negative affect; Positive Urgency, or the tendency to act rashly when experiencing positive affect; Lack of Premeditation or Planning, or a lack of planfulness and sufficient consideration of consequences of behavior; Lack of Perseverance, or difficulty in remaining focused on a task; and Sensation Seeking, or the tendency to seek novel and exciting experiences. Youths responded to each item on a 4-point Likert scale that ranged from 1 (very much like me) to 4 (not at all like me). For a description of the psychometric properties, see Cyders et al., 2007 [62], and Barch et al., 2018 [51].

## 8. Statistical Analysis

Aim 1: Extracting COI 2.0 Factors. We used R version 2021.09.1 and utilized the GPARotation and psych packages to conduct an exploratory factor analysis (EFA) of the 29 COI 2.0 indicators. We used a geomin (oblique, or correlated) rotation and a maximum likelihood estimator on the tetrachoric correlation matrix. To determine which model best described the data, we used parallel analysis, Velicer’s minimum average partial test (MAP), and the test of very simple structure (VSS).

Aim 2: Associations between COI 2.0 Factors and External Criteria. We used linear mixed-effects models to estimate the associations among COI 2.0 factors and youth psychopathology and allied traits (i.e., impulsivity, behavioral approach). We accounted for the nonindependence of participants within each study site and family using random effects. We further included fixed effects for age, sex at birth, and, in alignment with the recommendations of the ABCD Study [42,64,65], individual level of socioeconomic parent education and household income. Accounting for socioeconomic factors at the individual level (parent education and household income) allowed us to examine the independent role of neighborhood and environmental factors. See the supplement for information on the coding of parent education and household income.

## 9. Results

### 9.1. Aim 1

EFA supported a four-factor solution (Appendix A) with moderately correlated factors reflecting Socioeconomic Attainment, Neighborhood Enrichment, Child Education, and Poverty Level. We identified the indicators that loaded onto the factors above 0.40 or below −0.40. Socioeconomic Attainment captures aspects of neighborhood socioeconomic attainment including adult education attainment, college enrollment, health insurance coverage, high-skill employment, and median household income. Neighborhood Enrichment captures aspects of neighborhood enrichment including proximity to licensed-center-based care and high-quality-center-based care, access to green space, and walkability. Child Education encompasses neighborhood third-grade math and reading school average proficiency scores. Finally, the fourth factor appears to capture aspects of neighborhoods that have access to social and economic resources reflecting neighborhood poverty, including increased access to healthy food, lower housing vacancy rate, fewer households on public assistance, lower poverty rate, and fewer single-family households.

Thirteen indicators did not load onto any factor: AP Enrollment, Child Education Enrollment, High School Graduation Rate, School Poverty, Teacher Experience, Heat Exposure, Ozone, PM2.5, Hazardous Waste, Industrial Pollutants, Homeownership, Employment Rate, and Commute Duration. We treated these thirteen indicators as stand-alone neighborhood indicators and observed the associations with our outcomes (see Appendix A). We then excluded the indicators that did not load strongly on any factor and ran another EFA (see Table 2).

### 9.2. Aim 2

Socioeconomic Attainment. Socioeconomic Attainment was significantly negatively associated with the BAS Drive (standardized B = −0.053, pFDR < 0.001, ηp^2^ = 0.002), Fun Seeking (B = −0.031, pFDR = 0.037, ηp^2^ = 0.0008), and Reward Responsiveness (B = −0.057, pFDR < 0.001, ηp^2^ = 0.003) subscales. Socioeconomic Attainment was also significantly positively associated with the UPPS-P Sensation Seeking (B = 0.064, pFDR < 0.001, ηp^2^ = 0.003) and Lack of Planning (B = 0.037, pFDR = 0.012, ηp^2^ = 0.002) factors. Socioeconomic Attainment was not significantly associated with the ASEBA CBCL internalizing symptoms, externalizing symptoms, attention problems, or thought problems scales, the BIS composite, or the UPPS-P Positive Urgency, Negative Urgency, or Lack of Perseveration factors (see Figure 1).

Neighborhood Enrichment. Neighborhood Enrichment was significantly positively associated with the BAS subscale Drive (B = 0.049, pFDR < 0.001, ηp^2^ = 0.006) and significantly negatively associated with the CBCL internalizing symptoms scale (B = −0.039, pFDR = 0.011, ηp^2^ = 0.00362) and the UPPS-P Lack of Planning (B = −0.036, pFDR = 0.006, ηp^2^ = 0.03) factor. Neighborhood Enrichment was not significantly associated with the BAS Reward Responsiveness or Fun Seeking subscales, the CBCL externalizing symptoms, attention problems, or thought problems scales, the BIS composite, or the UPPS-P Positive Urgency, Negative Urgency, Sensation Seeking, or Lack of Perseveration factors (see Figure 2).

Child Education. Child Education was significantly negatively associated with the BAS Drive (B = −0.060, pFDR < 0.001, ηp^2^ = 0.003), Fun Seeking (B = −0.039, pFDR = 0.005, ηp^2^ = 0.002), and Reward Responsiveness (B = −0.047, pFDR < 0.001, ηp^2^ = 0.002) subscales. In contrast, Child Education was significantly positively associated with the UPPS-P Lack of Planning (B = 0.039, pFDR = 0.005, ηp^2^ = 0.003) factor. Child Education was not significantly associated with the CBCL internalizing symptoms, externalizing symptoms, attention problems, or thought problems scales, the BIS composite, or the UPPS-P Positive Urgency, Negative Urgency, Sensation Seeking, or Lack of Perseveration factors (see Figure 3).

Neighborhood Poverty. Neighborhood Poverty was significantly positively associated with the BAS Drive (B = 0.073, pFDR < 0.001, ηp^2^ = 0.005), Fun Seeking (B = 0.046, pFDR = 0.002, ηp^2^ = 0.003), and Reward Responsiveness (B = 0.048, pFDR = 0.001, ηp^2^ = 0.003) subscales and the UPPS-P Lack of Planning (B = −0.032, pFDR = 0.027, ηp^2^ = 0.004) factor. In contrast, Neighborhood Poverty was also significantly negatively associated with the CBCL internalizing symptoms (B = −0.038, pFDR = 0.016, ηp^2^ = 0.001) subscale and the UPPS-P Negative Urgency (B = 0.035, pFDR = 0.019, ηp^2^ = 0.002) and Positive Urgency (B = 0.037, pFDR = 0.014, ηp^2^ = 0.002) factors. Neighborhood Poverty factor scores were not significantly associated with the BIS composite, the CBCL externalizing symptoms, thought problems, or attention problems scales, or the UPPS-P Sensation Seeking and Lack of Perseveration scores (see Figure 4).

Stand-Alone Indicators: See the supplement for the associations between our stand-alone COI 2.0 indicators (that were not included in our factor scores) and the external validators (see Appendix A).

## 10. Discussion

We leveraged a data-driven approach with a large, diverse community sample of youths to illuminate the relationships between access to neighborhood resources and symptoms of psychopathology, impulsivity, and behavioral approach using data from the Adolescent Brain Cognitive Development Study. Using exploratory factor analysis (EFA), we identified four factors of child opportunity pertaining to (1) Child Education, (2) Socioeconomic Attainment, (3) Neighborhood Enrichment, and (4) Poverty Level. These neighborhood factors were differentially associated with psychopathology, impulsivity, and behavioral approach variables.

## 11. Factor Structure

We found that four factors best described the structure of the COI 2.0: Socioeconomic Attainment, Neighborhood Enrichment, Child Education, and Poverty Level. Socioeconomic Attainment may reflect neighborhood wealth and financial resources. Specifically, neighborhoods with higher financial resources may have more opportunity to have greater financial power to attract education opportunities, social networks, private businesses, and health care service providers [66,67,68,69]. Neighborhood Enrichment may represent aspects of the built environment that may promote enrichment and healthy development. Child Education may capture children’s early education access within schools, family settings, and communities (libraries, after-school programs, and youth/community programs), which may promote academic achievement^4^. Finally, our Poverty Level factor may be distinguished from Socioeconomic Attainment to represent more severe levels of poverty as these are aspects of the community that are likely to depend on local funding and amenities such as supermarkets, quality real estate and housing markets, access to transportation, and childcare support.

These factors align with the original three domains of the COI 2.0, education, social and economic, and health and environment [52], although we found dimensions that disentangled the social and economic factors into *attainment opportunities* and *poverty*. This finding is consistent with other studies that have found a three-factor solution of measures from the Americans’ Changing Lives (ACL) survey derived from the U.S. Census 68. Their three-factor solution included *ethnic and immigrant concentration*, *neighborhood disadvantage*, including measures of poverty rate, female-headed households, and receipt of public assistance, and *neighborhood affluence*, which included measures of adult educational attainment and employment [70]. Given that our study and other studies [70] have disentangled aspects of affluence or attainment and poverty, this may provide a more nuanced approach to disentangle individuals who have higher affluence or education/employment opportunities from individuals who do not require public assistance or do not meet the poverty threshold. It may also provide more information on targets specific to attainment opportunities and targets specific to poverty rate within communities that may be instrumental in harm reduction for downstream mental health problems. We were perhaps able to disentangle social and economic opportunities given the geographic variability associated with the ABCD Study.

To examine individual differences in psychopathology, we included both parent-reported youth psychopathology (e.g., internalizing and externalizing symptoms) and youth-reported personality (i.e., impulsivity, behavioral approach), which are robust and well-researched predictors of downstream mental health problems in adolescence and into young adulthood [62,71].

## 12. Cross-Sectional Links Between Neighborhood Opportunity and External Criterion

### 12.1. Behavioral Activation System

All four COI 2.0 factors were associated with aspects of the BAS—such that Socioeconomic Attainment and Child Education scores were negatively associated with the BAS Drive, Fun Seeking, and Reward Responsiveness subscales, whereas, inversely, Poverty Level scores were positively associated with the BAS subscales. The BAS subscales are related to reward-seeking behavior, including pursuing desired rewards and seeking out rewards spontaneously. Our findings are consistent with the previous literature suggesting that children who grow up in environments with less access to education and related socioeconomic opportunities may have differential mesolimbic and cortical structural development [72], which underlies the reward processing and sensitivity important for the BAS. Specifically, a study using the ABCD Study sample found that a high Area Deprivation Index, a composite of neighborhood disadvantage, including qualities of neighborhood poverty and financial opportunities, was linked with decreased recruitment of motivational neurocircuits, including the dorsal and ventral striatum, during reward anticipation [73]. Our results suggests that environments with fewer neighborhood resources (e.g., reduced access to healthy food, a greater number of housing vacancies, and a higher proportion of single-family households) cause less emotional reserve in the neighborhood, which may impact the levels of caregiving and community support available for youths and families, which may impact a youth’s willingness to pursue *risky* rewards during adolescence. These results add to the literature by utilizing the Child Opportunity Index 2.0, a neighborhood index that specifically caters to elements of healthy child and adolescent development. Furthermore, our results may provide multiple targets for public health policy and funding to aid the promotion of healthy child brain development, for example, enhancing community spaces, increasing the number of grocery stores available to neighborhoods, and increasing funding and public assistance, particularly for single-parent households. Future research should explore the longitudinal associations of child neighborhood opportunity and later cognition and behavior related to substance use and the mechanism and trajectory of how the neighborhood contributes to brain development.

Interestingly, Neighborhood Enrichment scores were significantly positively associated with the BAS Drive subscale, which differs from our previous findings and is the opposite of what we expected. This relationship may reflect that opportunities in the neighborhood that reflect physical activity levels (e.g., walkability, access to green spaces) and social opportunities with peers (e.g., community centers) are related to a higher drive for specific rewards, including high-energy activities and goal-oriented opportunities, such as engaging with peers or playing outside, rather than *risky* behaviors [74]. However, this is speculative, and future research should investigate the differential association of the BAS and differing rewarding activities in the neighborhood context.

### 12.2. UPPS-P

**Sensation Seeking.** Our findings revealed that Socioeconomic Attainment was positively associated with sensation seeking, or the pursuit of novel stimulation or thrilling experiences. While research has typically framed sensation seeking in the context of developmental risks such as substance use or risky sexual behavior [75], sensation seeking has been linked with psychological wellbeing and increased engagement in physical activity [76,77]. Other research using the ABCD Study suggests that involvement with extracurricular activities is positively related to measures of socioeconomic status (e.g., family income, caregiver education). Our findings may extend this literature and suggest that youths living in neighborhoods with higher socioeconomic attainment—or areas with a higher level of college enrollment and a higher number of people with high-skill employment—may have the opportunity for increased environmental exploration and the seeking of high-reward experiences. Future studies should disentangle levels of sensation seeking, impulsivity, and other resilience factors more generally in the ABCD Study sample to understand differential patterns of development in terms of positive outcomes in prospective and longitudinal studies.

**Lack of Planning.** Consistent with what we expected, we found that Neighborhood Enrichment scores were negatively associated with UPPS-P scores for Lack of Planning, or the tendency to act without thinking, an aspect of impulsivity. Our findings are consistent with the previous literature suggesting that adverse neighborhood conditions, such as exposure to crime or lower neighborhood socioeconomic status, are linked with lower self-control and inhibitory control among youths [78,79]. Higher levels of neighborhood engagement, such as more socialization within the neighborhood (e.g., access to green spaces and walkable neighborhoods, access to community and education centers) could be a mechanism to promote social cohesion, autonomy, work and future orientation, better school performance, or the general development of self-control and inhibitory control [79,80].

Interestingly, Socioeconomic Attainment and Child Education scores were positively associated with the UPPS-P subscale Lack of Planning. This finding is inconsistent with previous research, which suggested that aspects of the neighborhood (higher socioeconomic status of the neighborhood or school quality and support) promote better planning, organization, and overall executive functioning skills in youths [78,79,80,81,82,83]. However, it is possible that the youths in this sample who live in a more affluent neighborhood that offers more opportunities for structure and organization are not as autonomous as youths living in more disadvantaged neighborhoods. This is consistent with some research that suggests links between affluent neighborhoods and lower educational attainment and poorer mental health outcomes [84,85]. However, more research is needed to investigate the differential effects of neighborhood quality, including aspects of wealth and child education, on child development, and particularly to investigate the impact of neighborhood wealth on adolescent autonomy and its ability to foster independence.

Furthermore, we found increased Poverty Level scores were associated with a *decreased* lack of planning, i.e., youths living in more deprived neighborhoods behave less impulsively. This is consistent with previous research suggesting that individuals in a more resource-scarce context may have increased planning and skills [86,87,88]. One possible mechanism that could explain this relationship is that living in more resource-deprived neighborhoods, including more impoverished neighborhoods, may be linked with more neighborhood cohesion, which could be protective of more adaptive decision-making [89]. These findings may warrant further exploration to investigate child education and aspects of poverty in relation to lack of premeditation at different time points.

### 12.3. Positive and Negative Urgency

Additionally, Poverty Level scores were significantly positively associated with the UPPS-P Negative Urgency and Positive Urgency scales. Positive and negative urgency are aspects of impulsivity that relate to the tendency to act rashly while in an intense emotional state, either positive or negative [62,90]. To our knowledge, our study is the first to explore the link between neighborhood factors (e.g., neighborhood poverty) and urgency. However, our findings are similar to findings from another study using the ABCD Study at baseline which found that higher parental educational attainment was linked with lower levels of negative urgency in girls but not boys. Our findings suggest that growing up in environments that lack resources (e.g., lack caregiver support) and have a higher poverty rate is linked with higher impulsivity in youth, which is consistent overall with the literature investigating resource deprivation and impulsivity [91]. Future studies should investigate this relationship as youths age into adolescence and into adulthood to see the lasting effects of neighborhood resource deprivation.

### 12.4. Psychopathology

Neighborhood Enrichment scores were significantly negatively associated with the CBCL internalizing symptoms scale. This finding is consistent with other research that found that increased neighborhood connectedness related to lower levels of depressive symptomology [92]. Additionally, other studies have found that aspects of green space and urban environment, indicators included in the neighborhood enrichment factors such as walkability and availability child education centers, related to reduced depressive and anxiety symptoms and mental wellbeing among youths [93,94]. One possible mechanism of this is that neighborhood enrichment may offer areas for children and youths to gather, which may promote social opportunities and encourage physical activity [95,96,97], which, in turn, may reduce the risk of symptoms associated with depression [98,99]. Additionally, social connectedness has been cited as a protective factor in promoting stronger connections that promote positive mental health outcomes in adolescents [100,101]. A possible mechanism for this relationship could be the deprivation of resources at the neighborhood level, which is linked to the social and interpersonal networking opportunities that are more prevalent in more affluent neighborhoods. Our findings add to the existing literature and suggest that policies that target enhancing neighborhood enrichment opportunities and systems and legislation that reduces the exposure of children to various levels of poverty may be particularly advantageous in promoting the wellbeing of children as they age into adolescents.

## 13. Limitations

The Adolescent Brain Cognitive Development (ABCD) Study offers a unique opportunity to investigate the impact of environmental characteristics, both built and natural, and social environmental factors [42] on developing cognitive and brain health while utilizing multiple measurements of neighborhood disadvantage. However, there are several limitations that need to be discussed. First, geospatial data may have limited spatial resolution. COI 2.0 data were examined at the census tract level; however, there may be some measures that warrant smaller units (e.g., a census block) or larger units (e.g., a census track or county; see Cardenas-Iniguez et al., 2024 [42]). Geospatial data may also have limited temporal resolution. The ABCD Study baseline data were collected starting from 2016, but the COI 2.0 data were from 2015. Neighborhood characteristics may have undergone significant changes in the intervening period. The COI 3.0 (2024) has recently been released, with several methodological changes from the COI 2.0. While the COI 3.0 has since been released, linked data were not yet available for ABCD Curated Release 5.1. These updates include changes in data sourcing, use of AI and machine learning techniques to improve data quality, and the addition of several indicator and subdomain scores. Future studies may consider re-running analyses using the COI 3.0.

Second, we have a limited ability to estimate the extent of participants’ exposure to their neighborhood environment due to the cross-sectional nature of the analyses. Neighborhood quality was only analyzed for the primary residence at baseline. Participants lived at their primary residences for at least 80% of the time throughout the week. However, the sample includes individuals who may have lived at this address for as little as one month or for up to 100% of their lifetime [53,64]. Therefore, the potential cumulative effects of exposure to neighborhood environments are unknown. Third, we are unable to make causal inferences due to the cross-sectional design. Follow-up research utilizing a longitudinal design may better account for these developmental effects. Importantly, as expected, our results yielded relatively small effect sizes; however, previous studies have cited that, within the ABCD Study, small effect sizes are often reproducible and considered meaningful [102].

Fourth, although the ABCD Study aimed to recruit a socio-demographically representative sample of the U.S. population, the ABCD cohort is over-representative of economically advantaged, highly educated families. Furthermore, participants were limited to the 21 study sites, the selection of which was constrained by the need for expert research personnel and neuroimaging facilities [46]. As these requirements tend to be concentrated in metropolitan locations, the ABCD cohort may be under-representative of rural youths. Additionally, using environmental estimates and geocoded or census-linked data does not map directly onto the lived experience or subjective experience of exposures, which are necessary to consider when investigating the impact of environment on development and health outcomes. Thus, future studies should investigate the interaction of subjective experiences and geocoded estimates of environmental exposure. Furthermore, an important factor within a neighborhood and individual context is considering the transmission of intergeneration access and control over wealth. For example, families with less income have a higher likelihood to live intergenerationally in lower-income environments. While we considered individual levels of household income and parental education, which may be seen as a robust indicator of available household resources, we did not directly study intergenerational wealth or education. Thus, future studies should incorporate the transmission of wealth and access to resources within their research questions. Additionally, as our study was preliminary and primarily aimed to identify a factor structure of the COI 2.0 in the ABCD Study, we did not consider or control for intergenerational transmission of psychological states or cognition or levels of achievement, for example, the potential influence of parental or intergenerational mental health or cognition, on our outcomes. Future studies that investigate later waves of the study and that include perceived discrimination, intergenerational transmission of wealth, access to resources, psychological states, and other objective measures will be paramount in understanding the intersection of lived experience, intergenerational transmission of wealth and mental health, and neighborhood opportunity.

## 14. Conclusions

To our knowledge, this study was the first large, multisite study that disentangled the 29 indicators of neighborhood resources via the Child Opportunity Index 2.0, consolidated them into four broad dimensions (Socioeconomic Attainment, Child Education, Neighborhood Enrichment, and Poverty Level), and externally validated them with scores relating to psychopathology, impulsivity, and behavioral approach. Taken together, child mental health, aspects of impulsivity, behavioral activation, and externalizing and internalizing are associated with the neighborhood factors of Socioeconomic Attainment, Neighborhood Enrichment, Child Education, and Poverty Level. For example, Socioeconomic Attainment and Child Education were broadly related to a more reduced behavioral approach system, which may reflect a less impulsive profile. For example, reduced fun seeking, reward responsiveness, and drive may indicate a reduction in *thrilling-seeking* or *risky* decision-making [72]. Alternatively, our study found increased Poverty Level scores were broadly related to increased externalizing symptoms, an increased behavioral activation system, and increased aspects of impulsivity. This may reflect a more *behaviorally* impulsive profile. This study provides important implications, e.g., neighborhood opportunity promoting healthy development may be important to reward and inhibitory- and impulse-control systems in children over and above individual parental education and the income of the household. These reward systems may underlie important behavioral and cognitive development that leads to downstream risky behaviors including substance use behaviors and risky decision-making in youth. Future research should investigate a more nuanced approach and investigate the differential effects of factor scores on differing aspects of child psychopathology and impulsivity as well as explore this trajectory over time as children in the ABCD Study sample age into adolescence.

## Figures and Tables

**Figure 1 ijerph-22-00228-f001:**
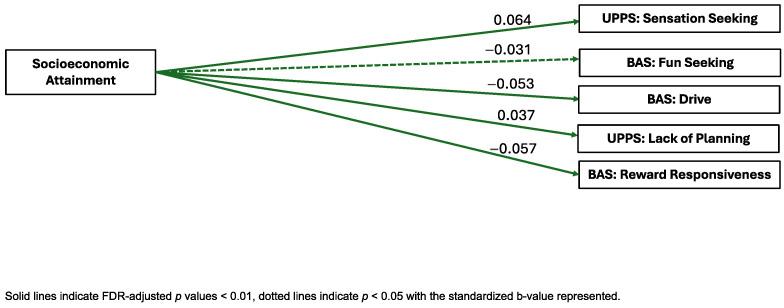
Visual representation of linear mixed-effects regressions of factor scores and external criterion.

**Figure 2 ijerph-22-00228-f002:**
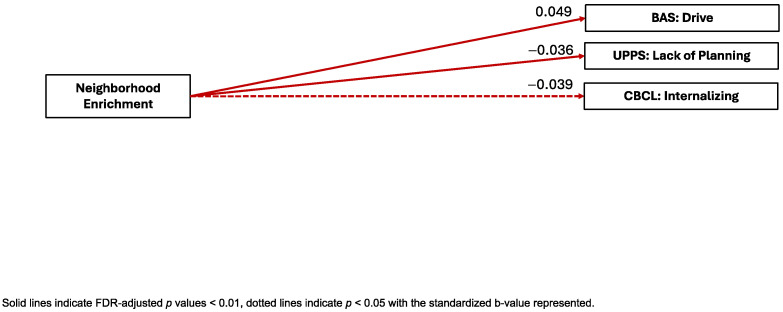
Visual representation of linear mixed-effects regressions of factor scores and external criterion.

**Figure 3 ijerph-22-00228-f003:**
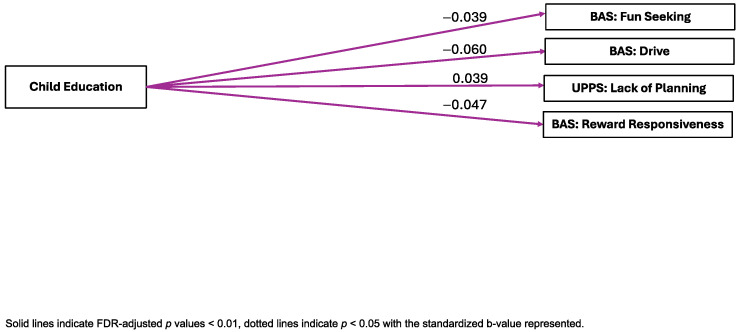
Visual representation of linear mixed-effects regressions of factor scores and external criterion.

**Figure 4 ijerph-22-00228-f004:**
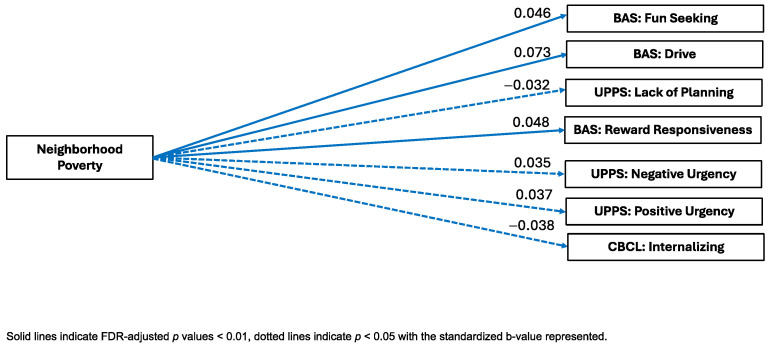
Visual representation of linear mixed-effects regressions of factor scores and external criterion.

**Table 1 ijerph-22-00228-t001:** Demographic characteristics.

	Total Sample (*n* = 9767)
Sex, *n* (%)	
Female	4652 (47.62)
Male	5113 (52.36)
Intersex Male	2 (0.02)
Age, months years, M (SD)	119.01 [9.92] (7.49)
Household Income, *n* (%)	
<USD 50K	2817 (28.84)
≥USD 50 and ≤100K	2785 (28.51))
≥USD 100K	4165 (42.64)
Parent Education, M (SD)	3.66 (1.05)
Race, *n* (%)	
White	5376 (55.04)
Black	1291 (13.22)
Asian	200 (2.05)
Hispanic	1868 (19.13)
Other	1032 (10.57)
BAS Subscales M (SD)	
Drive	4.05 (3.02)
Fun Seeking	5.69 (2.63)
Reward Responsiveness	10.98 (2.90)
BIS Sum M (SD)	9.50 (3.75)
UPPS-P Subscales, M (SD)	
Negative Urgency	8.49 (2.64)
Lack of Premeditation	7.76 (2.36)
Lack of Perseverance	7.03 (2.24)
Sensation Seeking	9.80 (2.68)
Positive Urgency	7.96 (2.94)
COI 2.0, M (SD)	
Total COI Summary Score	61.44 (30.16)
ASEBA Subscales, M (SD)	5.03 (5.44)
Internalizing Symptoms	4.37 (5.74)
Externalizing Symptoms	1.62 (2.17)
Thought Problems	2.95 (3.45)
Attention Problems	

BAS = Behavioral Activation Scale; BIS = Behavioral Inhibition Scale; UPPS-P = Urgency, Premeditation, Perseverance, Sensation Seeking, and Positive Urgency; COI = Childhood Opportunity Index; ASEBA = Achenbach System of Empirically Based Assessment.

**Table 2 ijerph-22-00228-t002:** Four-factor solution of the Childhood Opportunity Index 2.0.

Indicator	Child Education	Socioeconomic Attainment	Neighborhood Enrichment	Poverty Level
Education Attainment	0.10	0.87	0.07	−0.08
College Enrollment	0.09	0.47	0.08	0.33
Third-Grade Math	1.00	−0.01	0.00	0.00
Third-Grade Reading	0.93	0.03	−0.01	0.00
CEC	−0.07	0.17	0.84	0.04
High-Quality Child Education Centers	0.00	0.27	0.58	0.09
Healthy Food	0.01	0.16	−0.12	0.75
Green Space	−0.03	−0.19	0.85	−0.04
Health Insurance	−0.01	0.60	−0.23	−0.11
Housing Vacancy	−0.18	0.11	−0.09	0.59
Walkability	0.04	−0.02	0.81	0.00
Poverty Rate	−0.03	−0.10	0.11	0.79
Public Assistance	0.02	−0.20	0.05	0.79
High-Skill Employment	0.06	0.88	0.00	−0.09
Household Income	0.14	0.44	−0.04	−0.39
Single-Family Households	−0.12	−0.14	0.08	0.66
Factor Correlations				
Child Education	-	-	-	-
Socioeconomic Attainment	0.57	-	-	-
Neighborhood Enrichment	−0.24	−0.05	-	-
Poverty Level	−0.59	−0.57	0.33	-

Note: loading values are standardized.

## Data Availability

Data used in the preparation of this article were obtained from the Adolescent Brain Cognitive Development SM (ABCD) Study (https://abcdstudy.org, accessed on 4 February 2025), held in the NIMH Data Archive (NDA).

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
