# Peer review of "The Childhood Opportunity Index 2.0: Factor Structure in 9–10 Year Olds in the Adolescent Brain Cognitive Development Study"

_ijerph, 2025, doi:10.3390/ijerph22020228_

Round 1
Reviewer 1 Report
Comments and Suggestions for Authors
Review for the ijerph-3344221:
1. In the Abstract, this study noted a total of 10776 data points. However, in Table 1, the total sample size is N = 9767. Why are the numbers of samples inconsistent? The authors should provide detailed rationale for this. For example, approximately 10% participants were excluded, why they were excluded?
2. This study emphasized the “Adolescent Brain Cognitive Development” in the Title. The authors should elaborate more detailed information on ABCD in the Introduction section, otherwise the readers are not clear about the purpose of using the ABCD.
3. Since there was Childhood Opportunity Index (COI), the reason for conducting the Childhood Opportunity Index 2.0 should be discussed more extensively. By the way, the authors should discuss the distinction of these two in the DISCUSSION section.
4. It is not clear how the 29 indicators were measured. In the Supplementary materials, all the 29 indicators were provided; however, more information should be provided with respect to the measurement item of each indicator. Taking “Public Assistance” as an example, who reported this information, did the children report their environment’s public assistance? What is the specific item of this indicator?
5. In Table 2, why the factor loading of “Third Grade Math” is 1.00? Since this index falls within the same dimension as the other index (i.e., Third Grade Reading).
6. There are many obvious formatting errors in this study, some of which are listed below but not fully covered:
a. Line 35, there is an extra dot after the word "more";
b. Line 95, there is extra space before however;
c. Line 174, there is an “e.g.,” in parentheses, but there’s nothing after it. Why?
d. These three Figures (i.e., Figure 1.1, Figure 1.2, and Figure 1.3) look blurred.
The authors should check and modify carefully for similar formatting errors before uploading their manuscript.
Reviewer 2 Report
Comments and Suggestions for Authors
Thank you for allowing me to review this study. The researchers address the associations between built physical and social environment and child neural and cognitive development. The goal of the paper is interesting. However, it is unacceptable in its current conceptualization and interpretations and propositions.
The objective of this study was to identify the factor structure and correlates of 29 environmental, education, and socioeconomic indicators of neighborhood resources as measured by the Child Opportunity Index 2.0 (COI 2.0) in a sample of youth aged 9-10 enrolled in the Adolescent Brain Cognitive Development Study (ABCD).
This study used baseline data of the ABCD Study (n = 10,776, ages 9-10). They used structural equation modeling to investigate the factor structure of neighborhood variables (e.g., indicators of neighborhood quality including access to early child education, health insurance, walkability). They externally validated these factors with measures of psychopathology, impulsivity, and behavioral activation and inhibition. Exploratory factor analyses identified four factors: Neighborhood Enrichment, Socioeconomic Attainment, Child Education, and Poverty Level. Socioeconomic Attainment and Child Education were associated with overall reduced impulsivity and behavioral activation system whereas increased Poverty was associated with increased externalizing symptoms, increased behavioral activation system, and increased aspects of impulsivity. Distinct dimensions of neighborhood opportunity were differentially associated with aspects of psychopathology, impulsivity, and behavioral approach.
They suggest that neighborhood opportunity may have a unique impact on neurodevelopment and cognition.
While I appreciate the justification that pre-adolescents need a built environment with protective factors, I was concerned about their clinical insight on behavior disorders. My main concern in the abstract, introduction, and discussion is their suggestion that built environment “impacts” psychological development or states.
There are several facts that are gravely neglected:
a) Intergenerational transmission of access and control over wealth (people from less income family environments have a higher likelihood to live intergenerationally in less income environments)
b) Intergenerational transmission of psychological states (people from family environments with mental health problems have a higher likelihood to transmit this characteristic intergenerationally and live intergenerationally in less income environments/with less educated parents/less psychological opportunities, people from family environments with mental health problems have a higher likelihood to transmit this characteristic intergenerationally and live intergenerationally in less income environments/with less educated parents/less resilience and motivational opportunities)
c) Intergenerational transmission of achievement (people who are less educated have a higher likelihood to be from less educated parents, and there is a intergenerational transmission of achievement potential from neurodevelopmental characteristics that are related to both intelligence and attentional skills).
Therefore, to assume that changing the built environment would change that seems like quite a stretch.
Instead, by acknowledging these, they can say that the built environment can (a) create opportunities for socialization and human potential realization, and (b) contribute to resilience and put in some citations on psychological resilience and built environment or biophilic design.
The problems are in the introduction and discussion (literature review and conceptual justification in introduction and the contribution of built environment as a protective asset rather than a causal driver). T
Example, research suggests that the lack of neighborhood opportunity, socioeconomic disadvantage, or urban social stressors (e.g., urbanicity) may “lead to” increased sensitivity in physiological stress systems including the hypothalamic-pituitary adrenal (HPA) axis among youth and adults.
Also, reduced neighborhood opportunity may promote mental fatigue which may “lead to” the depletion of top-down direct or voluntary attentional control, poorer self-control, and response inhibition among you t
Therefore, while I think the correlations are interesting, the authors need to reconceptualize the object of study (built environment) can be a protective factor rather than a “driver” (which sounds unidirectional and even worse, causal). They can also say that these findings of built environment likely serve as a protective marker for other ages of vulnerability (emerging adulthood, elderly etc). It needs a bit more clinical reconceptualization by providing a more balanced and nuanced view of the objective and results.
Reviewer 3 Report
Comments and Suggestions for Authors
Excellent sample and methodology. Superb literature review. The 4 factors are all meaningful and clearly defined. The Instruments section needs work. Provide reliability and validity data for all scales and sub-scales that are used as criteria in this study. Provide information about normative samples for each instrument. The biggest problem that needs to be addressed concerns the use of statistical significance for interpreting the results of the study. With huge Ns, tiny coefficients are statistically significant. That does not make them practically meaningful. All values of B in the mixed regression model are about 0 plus or minus about .05. The authors need to identify a valid metric of effect size, and use that metric to interpret which differences should be interpreted (perhaps those that account for 2% or more of the variance). They also need to be extremely cautious when interpreting the differences between effect size. Right now, I am not sure it is really possible to understand these results, or interpret and discuss them within the context of the wealth of previous literature, based solely on statistical significance.
Round 2
Reviewer 3 Report
Comments and Suggestions for Authors
The authors have revised the article appropriately and I accept their rationale for why even small differences have merit.